# Food Security in Israel: Challenges and Policies

**DOI:** 10.3390/foods13020187

**Published:** 2024-01-06

**Authors:** Ayal Kimhi

**Affiliations:** 1Shoresh Institution for Socioeconomic Research, Kochav Yair 44864, Israel; ayal@shoresh.institute or ayal.kimhi@mail.huji.ac.il; 2Department of Environmental Economics and Management, Institute of Environmental Sciences, The Robert H. Smith Faculty of Agriculture, Food and Environment, The Hebrew University of Jerusalem, Rehovot 76100, Israel

**Keywords:** food security, Israel, triple risk, risk-management strategy

## Abstract

This article analyzes Israel’s food security in comparison to other developed countries, using multiple indicators divided into four sections: food availability, food affordability, food quality and safety, and natural resources and resilience. Overall, the state of food security in Israel is better than in most countries, but the threats to food security arising from the triple risk of climate change, international conflicts, and disruptions in global supply chains, require better preparation for the future. Israel’s population growth and the slowdown in the growth rate of its agricultural production, as well as the short-term political desire to reduce prices, are leading the country to increasingly rely on food imports. Such imports expose Israel to even greater global risks, and require the formulation of a risk-management strategy that will balance local production and imports. The global triple risk to food security is currently exacerbated for Israel by the risk of shortage of labor due to the security situation, making this risk-management strategy even more necessary. This calls for the establishment of a governmental authority to oversee the formulation of a long-term food-security strategy, to break it down into feasible objectives and policy measures, and to supervise their implementation. Most importantly, in order to maintain and perhaps even enhance the productive capacity of the agricultural sector, the government must reinstall trust between farmers and the state by establishing a stable long-term policy environment.

## 1. Introduction

The internationally accepted definition of food security is that all people, at all times, have physical, social, and economic access to sufficient, safe, and nutritious food that meets their food preferences and dietary needs for an active and healthy life [1]. Measuring the level of food security, at a global level, at a country level, at a household level, or at an individual level, is a non-negligible challenge, but all governments, and many NGOs, see themselves as having an obligation to ensure the maximum level of food security for the citizens of their country. Food security is an important component of national security, and its promotion requires a combination of policy measures in the fields of welfare, health, agriculture, environmental quality, and international trade. Modern food systems have increasingly succeeded in improving the access of individuals and households to food, in terms of both quantity and price. But, they have been less successful in improving the nutritional and health value of the food basket, in protecting the environment [2], and in allowing all population groups to benefit equally from the improvements [3].

In the first years following Israel’s independence, there was not enough food in the country to feed the rapidly growing population resulting from the massive waves of immigration. Consequently, the government had to implement food rationing. However, massive investments in agriculture bore fruit. In Israel’s first four decades, the quantity of agricultural output grew faster than the country’s population (Figure 1). However, as Figure 1 shows, the growth rate of agricultural output is slowing down gradually, as in many developed countries [4], and is now lower than the rate of population growth. The slowdown in the growth of agricultural output stems from multiple causes. One is the loss of the most fertile farmland to urbanization and the move to more marginal land. Another cause is the shortage of water. Israel now desalinates water, but the cost of desalinated water is way too high for many crops. Farms rely more heavily on treated water, which is cheaper, but it is not suitable for all crops and may lead to yield losses in some crops. Environmental and health regulation of the use of pesticides and herbicides has tightened over the years, adversely affecting both yields and profitability. In addition, government support of agriculture has been declining over the years, and government policy in general has become less favorable to the farm sector.

Despite that, since the 1950s, Israel has never again experienced food shortages. During the 1970s, food prices in Israel moved closely with the OECD average (Figure 2). The high food prices during the 1980s were due to the hyperinflation and economic instability that Israel experienced at that time [5]. Food prices declined during the 1990s following the stabilization plan of 1985. The next time food prices entered the public discourse was in 2007–2008, when the prices of agricultural commodities in international markets soared and led to a significant increase in the price of food in Israel as well (Figure 2). The real increase in Israeli food prices was higher than the average real increase in the price of food in OECD countries. This contributed to the outbreak of social protests over the high cost-of-living in Israel in the summer of 2011. 

The Commission for Economic and Social Change (the “Trachtenberg Commission”) was established following the social protests. Its report listed the food sector among the sectors needing a price-reduction policy. At the same time, an “Inter-ministerial Commission for Examining the Level of Competitiveness and Prices in the Markets for Food and Consumer Products” (“Kedmi Committee”) was established to study the characteristics of the food- and consumer-products markets, to locate market failures, if any, and to formulate recommendations for improving consumer welfare.

The Kedmi committee report [6] placed most of the responsibility for the sharp increase in food prices that began in 2005 on the low competitiveness of the food sector. Competitiveness in the food sector declined significantly in 2005 with the purchase of the Clubmarket, the third-largest retail chain, by Shufersal, which resulted in an increase in the market share of the two largest chains (Shufersal and Mega). Market concentration allowed the large chains to raise food prices when producer prices rose due to the increase in the price of agricultural inputs, and to not lower food prices when producer prices fell (asymmetric price transmission) so that food prices remained high even after 2008, when the global recession led to a sharp drop in input prices and the exchange rate also fell, which should have made imported food cheaper. The committee’s policy recommendations focused on reducing import tariffs on imported food products and actions in the consumer sector, as well as actions to reduce market concentration. The bulk of these recommendations were not implemented.

Discourse on food security followed the social protests focused on food prices. However, disruptions in global supply chains following the outbreak of the COVID pandemic in 2020 turned the spotlight on the danger of possible food shortages in Israel. For the first time in decades, a serious public discussion began on the limitations of globalization and the importance of self-production of basic consumer goods. This discussion intensified with the onset of Russia’s war on Ukraine, as both countries are large exporters of both energy and food. The disruptions to these countries’ exports caused global price increases of agricultural commodities such as wheat and sunflower oil, as well as various energy products. Since modern agriculture needs a considerable amount of energy for its production and transport, higher energy prices lead to food-price increases. Finally, the onset of Israel’s war against the Hamas terrorist organization has led to offensive verbal responses from leaders of neighboring Islamic countries that are important sources of imported produce to Israel, causing much concern in Israel about the long-term reliability of imports from these countries.

In light of the emerging threats to food security in Israel, several key questions arise:How does Israel’s food security compare internationally?To what extent can (or should) Israel rely on self-production of its food?To what extent does the rise in food prices, and especially the prices of fresh agricultural produce, pose a threat to food security?How successful are existing policy measures in dealing with these threats, and what is the potential of other policy measures?

This study addresses these questions, primarily within the context of a multi-indicator food-security index. The previous literature mostly dealt with a narrower list of indicators. Endeweld and Silber [7] examined the historical development of the supply of food at the macro level and also the nutritional insecurity at the micro level. Griver and Fischhendler [8] show that Israeli food-security policy changed focus over the years following global and local events. Tal [9] discusses the role of public investments in research, development, and extension of enhancing food security in Israel. None of these authors offered an overall inclusive discussion on food security in Israel, especially not the future threats and the role of contemporary policy changes. This paper also discusses the unique situation of Israeli agriculture in light of the current armed conflict, and highlights the need for a long-term risk-management strategy for food security.

## 2. Methodology

This study utilizes an index of food security developed by the Economist Group. The index is constructed as a weighted average of a long list of elements affecting food security at a national level [10]. This index, compiled with the help of a team of experts, is divided into four main sections: food availability, food affordability, food quality and safety, and natural resources and resilience. Each of these sections includes a variable number of measures (Table 1), and the overall food-security index is a weighted average of all measures.

## 3. Food Security in Israel Compared to Other Countries

Figure 3 shows the food-security index of all countries for which the data are available as a function of GDP per capita. Not surprisingly, food security increases as GDP per capita increases. Specifically, the food-security index increases faster at lower levels of GDP per capita, i.e., among the poorest countries. The growth in the index declines as countries become wealthier. The solid line in the figure indicates a polynomial fit, and shows that Israel’s food security is slightly better than what its GDP per capita predicts. Israel’s food security is ranked in 12th place among the 32 OECD countries that participated in the ranking (the country ranked first enjoys the highest food security). However, a closer look at the components of the index provides a slightly less-optimistic picture about the future, as it pertains to food security in Israel. In terms of food availability and affordability, Israel ranks fifth and seventh, respectively. In terms of food quality and safety, Israel ranks tenth. These rankings represent the state of food security in Israel today. 

However, in the field of natural resources and resilience, Israel is ranked second to last. This implies that if appropriate measures are not taken, food security in Israel is expected to deteriorate in the future. To gain a better understanding about what these rankings mean, this study delves into each of these components separately. 

### 3.1. Food Availability

The food availability index includes components of sufficiency of supply, agricultural R&D, agricultural infrastructure, volatility of agricultural production, political and social barriers to access, food loss, and policy commitment (Table 1). Israel’s high ranking in the field of food availability is primarily due to the fact that the country’s current food supply is able to satisfy the energy requirements (in terms of calories) of the population and much more (Each person’s energy requirement is the minimum number of calories they need to receive from food in order to fully function and to have an active immune system). However, in the area of agricultural R&D, which is intended to advance future food security, Israel receives much lower scores. On the one hand, public expenditure on agricultural R&D relative to the total agricultural product is about 42% of the total government investment as a share of GDP, which places Israel in the center of the distribution of the OECD countries (15th place among 32 countries). On the other hand, total-factor productivity (TFP) in Israel’s agriculture sector increased by only 3.4% between 2012–2021, one of the lowest growth rates in the OECD. An increase in TFP reflects the ability to increase output without changing the quantities of inputs, while indirectly reflecting the contribution of agricultural R&D, which in Israel’s case is relatively modest. (It should be noted that there is an academic debate about the methodology used to calculate TFP changes and the adaptation of this measure to the unique conditions of Israeli agriculture.)

Israel also ranks low in indicators of infrastructure quality. It is ranked at the bottom of the list of OECD countries in the areas of transport infrastructure, including roads, railways, and ports. On the other hand, it receives a high score in the field of irrigation infrastructure, since nearly half of its cultivated agricultural land is connected to an irrigation network. Israel ranks relatively high in the area of volatility in agricultural production. The reason for this may be the high percentage of land with an irrigation infrastructure, which makes crops less vulnerable to fluctuations in precipitation. 

The rating of Israeli agriculture in terms of political and social barriers is quite low. This stems from the danger of armed conflict, political instability, corruption, and gender inequality. In the area of commitment to a food-security policy, Israel is ranked in the upper part of the distribution, apparently due to the existence of a national food-security council. However, it is ranked in the lower part of the distribution in the area of food-security strategy, probably because no such strategy exists in Israel [12].

As in other developed countries, about a third of the food produced in Israel does not reach the plate. Food loss occurs throughout the supply chain, but mainly at both ends: in consumers’ homes and in agricultural farms [13]. However, food loss in Israel is not particularly high in comparison with other developed countries—with Israel ranking 9th in the OECD.

Ironically, food loss in consumers’ homes might have been lower if food prices were higher. Consumers purchase food in larger quantities than necessary because there is uncertainty about the amount of food they will need. Increased food purchases can be viewed as “insurance” against a greater-than-expected demand for food. The cost of insurance is the cost of the food that is ultimately thrown away, and the higher the price of food, the less insurance consumers will be willing to “buy”. 

Another facet of this phenomenon is food that is thrown away by institutional consumers, such as banquet halls. Loss of food in agricultural farms is also to a large extent a result of the low prices of agricultural produce. When the price is too low, a farmer may make a decision (which is completely right for him) not to harvest the crop and thus save the cost of harvesting, sorting, and transportation. It follows that striving for zero food loss is impractical [14].

However, the decisions, which may be correct from the point of view of individual consumers and producers, are not necessarily the right ones at a national and global level. This is because the production of food, which is incompletely consumed, has environmental costs that are not taken into account by private agents (producers and consumers) [15]. Hence, countries have an interest in reducing food loss, often doing so through civil society organizations, which deal with saving agricultural produce and transferring it to those in need. However, such actions also have side effects. For example, distributing surplus food to the needy is likely to reduce their food purchases, which will lead to lower agricultural produce prices, and, as a result, larger quantities of crops may not be harvested.

### 3.2. Food Affordability

Food availability is measured using the change in the cost of food, the poverty rate, an inequality-adjusted income index, and import tariffs (Table 1). As shown in Figure 2, food prices in Israel rose during the 1980s (a period of three-digit inflation and a major debt crisis in agriculture, which drove many farms out of business), relative to food prices in other developed countries. This trend reversed in the 1990s, with food prices in Israel falling sharply during the decade. Food prices rose again in the following decade, and especially in the years 2005–2008, when world food prices rose substantially.

However, the increase in Israeli food prices was above and beyond the price increases in most OECD countries. An underlying reason for this may be the “Shufersal” chain’s purchase of the “Clubmarket” retail chain, which was the third largest at the time. This purchase greatly increased the concentration on the food-retail sector. Since 2008, food prices in Israel have remained more or less stable. Overall, food prices in Israel increased since the end of the 1990s much more than in the OECD. It is possible that the strengthening of the Israeli currency (shekel) had a contribution to this, since the proportion of imports in the food basket of Israel is relatively high. Ben-David and Kimhi [16] found that compared to the total-consumption basket, food prices in Israel were about 3% lower than the OECD average in 2005, while in 2017 they were about 3% higher.

Prices of fruits and vegetables have increased in recent years compared to other food items. Figure 4 shows that the prices of fruits and vegetables moved closely with the price of the total food basket between 2000 and 2014. The steep rise in food prices during 2005–2008 reflects the trend in global food prices during that time. Since then, food prices experienced a modest rise over the years. However, starting in 2015, there was a steep increase in the prices of fruits and vegetables relative to the total food basket, reaching a maximum of 30% increase in 2020 compared to 2000. There are multiple reasons for this price rise, including a higher cost of production, tighter environmental regulations, larger produce variety, extensions of the cropping season, and, of course, climate-induced losses of output. The findings of Ben-David and Kimhi [16] indicate that this phenomenon is not unique to Israel. Fruits and vegetables became more expensive during this period in other developed countries as well. In fact, Ben-David and Kimhi [16] showed that prices of fruits and vegetables in Israel are lower than the average prices in the OECD countries when compared to the total consumption basket, both in 2005 and 2017. In addition, they showed that while the median wage in Israel enables the purchase of 15% fewer standard food baskets than the median wage in the OECD countries, it allows the purchase of 21% more fruit and vegetable baskets.

The fact that fruits and vegetables are more affordable in Israel than in other developed countries is perhaps relevant to the question of whether the removal of import barriers will succeed in reducing prices. Nonetheless, their price increase in recent years has been a concern for Israeli consumers. Fruits and vegetables are a significant component of the healthy food basket recommended by the Ministry of Health [17]. As such, their price increases hinder the recommended change in dietary habits. This conclusion is supported by Figure 5, which shows that the local per capita supply of fruits and vegetables in Israel has been declining since 2005. However, Israel’s ranking in the food-security index places it in second place in the area of food prices. This is because food prices in Israel have increased at a moderate rate relative to most OECD countries since 2010. Specifically, food prices in Israel increased by 1.6% between 2010 and 2021, while the average price increase in the OECD countries was 2.4%. Conversely, import tariffs on agricultural products place Israel in 27th place in the OECD in this area, since they are seen as a factor that makes food more expensive.

Israel’s poverty rates place it in 20th place among the OECD countries, and 27th in the area of income per capita adjusted for inequality (Table 1; Per capita income at 2011 prices according to purchasing power parity, adjusted for inequality according to the methodology in [10], which means that the greater the inequality, the smaller the adjusted income). Since Israel is one of the least equally developed countries [18], the concern for the public’s ability to purchase a basket of healthy food is focused on the weaker populations. Azarieva and others [19] showed that in Israel, as in any other country, the share of food expenditures out of total household expenses rises as incomes fall. Specifically, 42% of the total expenses of the lowest income quintile (the fifth of all households with the lowest income) are spent on food (Figure 6). 

In the second quintile from the bottom, only 22% of expenses are devoted to food, and this proportion continues to decrease as one moves up the income distribution. If the households in the lowest quintile were to consume a healthy food basket as defined by the Ministry of Health [17], they would have to spend almost two thirds of their total expenses on it, which is unrealistic. (The cost of a healthy food basket is defined as the cheapest way to purchase a food basket assembled according to the recommendations of the Ministry of Health.) Even in the second and third quintiles from the bottom, the cost of a basket of healthy food is higher than the basket of food actually bought.

These findings turn the spotlight towards the prices of healthy food items. Figure 7 shows Israel’s rank among 39 OECD countries in terms of the average daily costs of standard food baskets (PPP dollars) in 2017. These costs are based on the cheapest available food items in each country. The three baskets are the standard food basket that meets the caloric needs of the population (2330 kilo-calories), a basket that provides a nutrient-adequate diet, and a basket that provides a healthy diet (A healthy diet provides not only adequate calories but also adequate levels of all essential nutrients and food groups needed for an active and healthy life. The cost of a healthy diet is defined as the cost of the least expensive locally available foods to meet requirements for energy- and food-based dietary guidelines for a representative person within an energy balance at 2,330 kcal/day. The guidelines explicitly recommend food quantities for each food group and provide a wide regional representation [20]). In terms of the prices of a standard food basket, Israel is ranked 16th, above the medial OECD country, indicating that the cost of a standard food basket is higher than in the majority of OECD countries. When comparing the prices of a food basket that provides all the necessary nutrients, Israel is ranked in 32nd place, implying that the prices in Israel are lower than in most OECD countries. This is also the case when comparing the prices of a healthy food basket, as defined by the World Bank, where Israel is ranked 35th, close to the bottom of the distribution. In addition, between 2017 and 2020, the price of the healthy food basket in Israel increased by a little more than 2%, much less than in most developed countries (Figure 8). This implies that the healthy-food-affordability problem of the weaker households is not a problem of local price inflation. Rather, it reflects a global problem.

In this context, it is important to note that for decades Israel has been implementing price controls on products considered basic so that poorer populations will be able to purchase them at reasonable prices. The list of price-controlled products includes bread, salt, dairy products, and eggs [19]. Of the various bread products, price controls are applied to dark bread, white bread, and challah, products that are not considered particularly healthy. In contrast, the healthy food basket includes whole-wheat bread, whose price is double or more than the price of controlled bread, while its production cost is not much higher [21]. The price of salt is also controlled, which may lead to excess consumption with its adverse health consequences. The control over the prices of milk products and eggs is related to the planning policy of the milk- and egg-production system. However, the milk products whose prices are controlled include butter, cream, and hard cheeses which are high in fat, and their consumption in large quantities is not recommended by health professionals. In general, it can be said that the food-price-control policy does not coincide with the promotion of the consumption of a healthy food basket.

Another way to support a healthy diet among vulnerable populations is through direct aid. Civil society organizations operate several programs to supply food directly to needy households, some of which receive government support. The flagship program is the “Food Security Initiative”, under which each family receives monthly assistance of three types: a magnetic card worth NIS 250 for use in food chains (without the option of purchasing alcohol or tobacco), fruits and vegetables worth NIS 125, and “dry” food products worth NIS 125. According to the findings of the National Insurance Institute [22], in 2021 there were about 265,000 families in Israel, constituting roughly 8% of the population, that suffered from considerable food insecurity. Of these, only about 11,000 were supported by the food-security initiative.

### 3.3. Quality and Safety

The quality and safety index includes the subcomponents of dietary diversity, nutritional standards, micronutrient availability, protein quality, and food safety (Table 1). Israel is at the top of the list of developed countries in the areas of protein quality, micronutrient availability, and food safety (Table 1). The areas that slightly lower Israel’s position in the field of quality and safety are the low dietary-diversity (24th place) and nutritional standards, which places Israel in the center of the distribution of the OECD countries (The score in the area of nutritional standard is based on four components: Has the government issued guidelines or managed a public program to encourage a balanced diet? Does the government have a national plan to improve nutrition? Does the government require nutritional labeling on food-product packaging? And, does the government monitor the nutritional status of the population?).

### 3.4. Natural Resources and Resilience

The natural-resources and resilience index includes indices of exposure to climate change, the risk to water quantity and quality, changes in land and aquatic resources, sensitivity to import conditions and natural capital, political commitment to adaptation, and demographic stress (Table 1). Israel is suffering from the depletion of the natural resources needed to produce food, especially land and water. The agricultural sector is gradually losing farmland, especially quality land in the central region, in favor of other land uses such as housing and non-agricultural businesses. Also, the natural water resources that used to be available for agriculture are increasingly either polluted or being taken for other uses, while the desalinated water replacing them is much more expensive (On the other hand, increasingly, treated sewage water is available for agriculture, but this water is not suitable for all crops, and its cost to farmers is at the center of an intense public debate that has not yet been decided). The root of the problem, both in terms of land and water availability, is Israel’s rapid population growth and its increasing population density, both of which are unique to developed countries [23].

The problem of the resilience of the food supply in Israel to risks is derived from three main types of risk. One is due to climate change, which is expected to lead to an increase in temperatures, a decrease in precipitation, and above all, increasingly irregular extreme-weather events that impair local food production. The eastern Mediterranean region that Israel belongs to is considered one of the regions where the impact of climate change is expected to be the most severe.

However, when compared to other developed countries, Israel’s situation in the area of exposure to climate change is not particularly bad. It is ranked ninth in the OECD in this area; the risk of drought is particularly high, while the risk of flooding is particularly low. The risks of temperature increases and sea levels rising place Israel in the 10th and 14th places in the OECD, respectively (Table 1). Even in a global ranking of nearly 200 countries, Israel is less vulnerable to climate change than most (Figure 9). In contrast, Israel’s ranking in the field of readiness to deal with climate change is much lower. Moreover, while Israel’s ranking in the field of vulnerability is relatively stable and ranges from 13 to 15 in the period 1995–2020, the ranking in the field of readiness has been steadily declining from the 28th position in 1995 to 41st in 2020. (The vulnerability index represents objective conditions, such as climate change, over which the state has no influence. The readiness index reflects the actions taken by the state in order to deal with the challenges.)

Another type of risk arises from the growing dependence of agricultural production on energy products, most of which are imported to Israel, and whose prices are subject to considerable volatility. The third type of risk lies in the prices of imported food, which are affected by climate change, by supply chain disruptions—such as the one that occurred as a result of the COVID pandemic—and by violent conflicts damaging global food supply, such as Russia’s invasion of Ukraine [24]. In this context, it should be noted that Israel imports almost half of its food supply, and if one also adds the import of animal feed, which is necessary for the local production of meat, milk, and eggs, then Israel imports much more than half of its food supply. Although supply sensitivity contributes to only 10% of the general-resilience index, the dependence on food imports places Israel in last place in the OECD in this area.

**Figure 9 foods-13-00187-f009:**
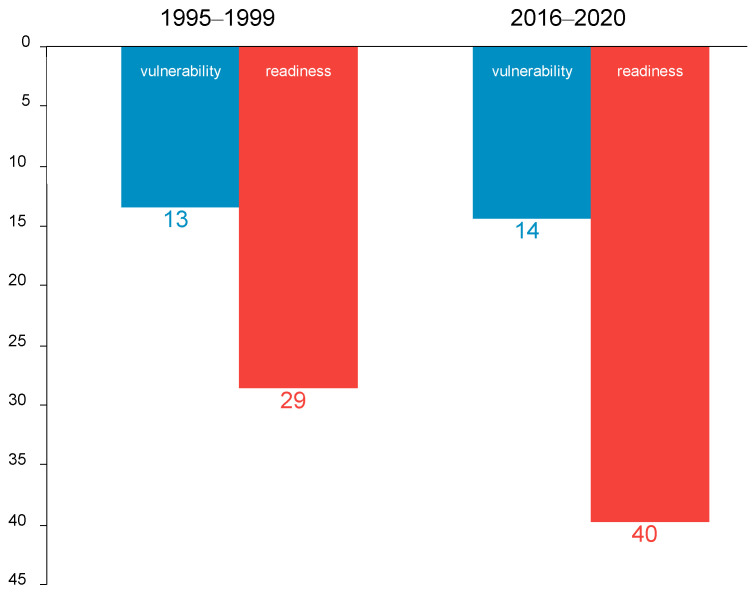
Ranking of Israel in terms of vulnerability and readiness to climate change (the highest the rank, the less vulnerable and the greater the readiness). Source: ND-GAIN [25]. This figure is adapted with permission from Ayal Kimhi (2022). Copyright 2022; Copyright Ayal Kimhi; Source: ND-GAIN (2022).

Figure 10 shows the percentage of self-sufficiency of the main food groups. It can be seen that Israel supplies itself with the lion’s share of fresh agricultural produce (the items in the box), while in other food products, which are responsible for a larger share of its caloric supply, the country relies mainly on imports. A calculation based on Figure 10 shows that 56% of Israel’s caloric supply relies on imports, and hence, the country’s considerable exposure to the risks emanating from global price fluctuations. Regarding the fresh agricultural produce, the high percentage of self-supply is largely rooted in the import restrictions and tariffs. The implementation of import reforms, a political issue that is continuously debated, may lead to a reduction in produce prices in the short term, but will certainly increase the exposure of Israel’s food supply to global risks.

Israel’s population is expected to double by 2065 [26]. The rate of increase in agricultural output has been on a downward trend for decades (Figure 1), and given the continued shrinking of cultivated-land areas in favor of urbanization and other land uses such as solar energy, no significant change is expected in this trend if Israel follows a business-as-usual strategy. This implies that the supply of food in Israel will be forced to rely increasingly on imports, with all the attendant risks. In order to preserve the local-production component of food, there will be a need for a significant technological advancement in agriculture that will make it possible to produce more output with fewer inputs, and at a reasonable cost. The next section delves more deeply into the role of agriculture in insuring future food security in Israel.

## 4. The Role of Agriculture

The growing reliance on food imports in general and the import of fresh agricultural produce in particular, does not diminish, and perhaps even increases, the importance of local agriculture as an important component of Israel’s food supply. In recent decades, there has been a noticeable slowdown in the growth of total-factor productivity in global agriculture (An increase in total-factor productivity reflects the increase that would have occurred in the quantity of output had the quantities of the factors of production not changed), both due to a decrease in public funding of research and development, and due to the effects of climate change which are already reflected in damage to crops [27].

In addition, the volatility of prices and supply in world markets is rising as a result of both climate change and crises such as the COVID pandemic and Russia’s war in Ukraine [28]. This implies that—especially during this period—it is important for the optimal portfolio of the food basket to include a component of locally produced food, in order to minimize the risk of shortage or a price hike. Toporov et al. [29] concluded that Israel is technically capable of producing food on its own that will satisfy most of the nutritional needs of its residents. However, it is not clear what the cost of such an autarkic policy would be, and whether it is possible to change consumers’ feeding habits accordingly, so this conclusion is not particularly relevant. The more relevant question is of which crops can be grown in Israel at a reasonable cost, taking into account both the current alternatives and the future risks? Amdur [30] examined a limited number of agricultural products and found that most of them are imported or can be imported from countries subject to a higher climate risk than Israel, which calls into question the ability to rely on imports of these products in the long term.

Maintaining the local production capacity of fresh agricultural produce requires an adequate income for farmers. Figure 11 shows that, since the 1980s, prices of agricultural inputs have risen faster than the price of output. If agriculture had not benefited from technological advancements and improvements in productivity during this period, it could be concluded that over the years the profitability of agriculture gradually eroded. Improvements in productivity without price changes would have resulted in an increase in profitability. In the absence of a reliable measure of the profitability of agriculture, the fact that many farms have stopped their productive activities over the years suggests that profitability has fallen. At the same time, the gap between the price of food for consumers and the price received by farmers has been widening, so that even during periods when the price of food increased, farmers did not necessarily benefit from this.

The decrease in the number of farms is reflected in the decrease in the number of self-employed persons whose main source of income is agriculture (Many farmers also make a living from other gainful activities, either because the income from agriculture is not sufficient or in order to diversify the sources of income due to the large volatility of agricultural income). Figure 12 shows that the number of farmers declined considerably over the years, from about 75,000 in 1960 to about 12,000 in 2020, a decrease of more than 80 percent. The decline was not uniform though. In fact, it can be seen that the rate of decrease in the number of farmers is temporally correlated with the rate of decrease in agricultural output prices, which fell by more than half in real terms during the same period. In the 1970s, output prices stabilized (thanks to the opening of export markets), and at the same time there was a slowdown in the rate of decline in the number of self-employed farmers. The accelerated decline in the prices of agricultural output in the 1980s and 1990s, due to trade liberalization and increased competition, was accompanied by the acceleration of farmers leaving the field. However, the change in the trend of the prices of agricultural output from negative to positive that occurred in the last two decades did not stop the trend of farmers leaving, since the prices of agricultural inputs rose more than the prices of output (Figure 11), suggesting that profitability continued to erode, at least for small and less-productive farms (The agricultural sector consists of large farms (mainly in Kibbutzim) and smaller family farms (mainly in Moshavim). The Kibbutzim (collective farms) did not abandon agriculture even if the number of Kibbutz members involved in agriculture is small. In the family farm sector in the Moshav (cooperative village), on the other hand, a decrease in the number of self-employed farmers implies the exit of the farm from production, and this exit is almost always irreversible. Of course, when small farms exit the sector, other farms have access to more inputs (land and water) and can increase production. This can enhance their own profitability, even when overall profitability in agriculture declines).

The meaning of the decrease in the number of farms at the same time as the increase in agricultural output is that the size (in terms of production capacity) of the average farm increased, and even increased greatly. On the one hand, larger farms can become more efficient by exploiting economies of scale, thereby contributing to food security. On the other hand, the decreasing number of farms may increase the instability of the food supply. For example, it is enough for a number of large farms to stop producing to create an unexpected shortage of certain crops. From this, it follows that it is desirable to create a balance between the need to increase agricultural output and the need to preserve small- and medium-sized family farms. Studies have shown that concentration in a certain industry may have negative consequences not only for competitiveness, but also for innovation in the industry, and on the environment, health, and animal welfare [32].

Increasing investments in agricultural R&D and advanced mechanization may help increase agricultural output and the resilience of the food system. Indeed, one of the components of the agricultural reform proposed by the Ministry of Finance and the Ministry of Agriculture is an increased budget for R&D and capital investments. However, another significant component of the reform is a gradual (over five years) exposure of many crops to competing imports. It is doubtful whether the increase in R&D budgets will bear fruit in such a short period of time, so that only those farms managing to survive will benefit from it, and it is not clear how many of these there will be. Assuming that it will indeed be possible to import certain fruits and vegetables from nearby countries (mainly Turkey, Jordan, and Egypt) at low prices, the local production of those fruits and vegetables will certainly decrease, thereby accelerating the exit, especially among small family farms, from agriculture.

In the long term, the danger inherent in such a scenario is threefold. Farms in those neighboring countries are much less technologically advanced than their Israeli counterparts. Their access to irrigation water during periods of droughts is much more limited, compared to Israeli farms who can use desalinated water. Hence, they are much less prepared to deal with climate change than Israel [33], so the possibility of importing from them at low prices could very well diminish over the years. In addition to this, the political instability in these countries, and the fluctuations in their diplomatic relations with Israel, as recently reflected in the response of their leaders to Israel’s war against the Hamas terrorist organization, endanger the regular supply of agricultural products from them. Finally, it is not clear as to what extent the quality of the produce imported from these countries and the environmental and health standards of their farmers can be effectively monitored. The bottom line is that even if the reform will lead to cheaper fruits and vegetables in the short term, it endangers food security in Israel in the long term.

Israel has recently encountered another food-security threat. Israeli farms, and in particular fruit and vegetable farms, rely heavily on the services of foreign workers. Palestinians started working as daily laborers in agriculture, construction, and services since the late 1960s [34], but the worsening security situation during the Palestinian uprising of the late 1980s did not allow them to arrive regularly at work [35], and as a result, the government allowed employers to replace them with foreign workers that came for a five-year period each. The employment of Thai workers in agriculture started in 1993 in small numbers, but those numbers increased considerably in subsequent years [36]. Despite the efforts to limit and even reduce the number of Thai workers in subsequent years, motivated by the populist argument that they take the jobs of local unskilled workers, their numbers remained relatively stable, and agriculture became practically dependent on them [37]. As Figure 13 shows, foreign workers comprise more than a fourth of all agricultural workers in Israel and more than a third of the hired labor force.

The atrocities of the Hamas attack on 7 October 2023 and the subsequent war disrupted food security in Israel. On top of the loss of farmland, infrastructure, farm buildings, equipment, and livestock in the agricultural regions that were directly affected, the scores of people murdered, wounded, or kidnapped included foreign agricultural workers. As a result, as many as a third of all Thai workers, not only from the combat zones but from the entire country, left the country immediately. In addition, Palestinians who used to work in agriculture (and other sectors) were banned from entering the country. The immediate and unexpected loss of farm labor implied a threat to food security in both the short and the medium range. Farmers are using volunteers for harvesting this season’s crops, but this arrangement cannot continue much longer. Many farmers are hesitating to prepare the fields for next season because they expect that the shortage of labor will sustain. It is likely, then, that the supply of fresh agricultural products will decline in the medium range. The slack could be closed with imports, of course, but those farmers who will stop producing will find it more difficult to resume production in the future because of cash-flow problems. This highlights the importance of a risk-management strategy that will balance local production and imports, so that imports can be augmented when local production is insufficient and local production can be expanded when imports become expensive.

## 5. Discussion

This paper analyzed the food-security situation in Israel, highlighted the main concerns, and discussed the relevant policy responses. Food security in Israel seems to be satisfactory overall compared to other OECD countries; although, Israel is ranked at the bottom of the list in several areas, most notably resilience to future threats. The intensifying threats of climate change [38], international conflicts, and disruptions in the global supply chain require greater attention from policymakers with an eye to the future. Israel’s rapid population growth, which is expected to continue into the foreseeable future, and the slowdown in the rate of growth of its agricultural production over the years, indicate that Israel’s reliance on food imports will continue to increase. Importing food exposes Israel even more to global risks, and requires the formulation of a risk-management strategy. Such a strategy must include strengthening local production, especially in products where Israel does not have a significant relative disadvantage. Another threat that requires more attention from policy makers is the combined adverse impact of high and increasing prices of fresh produce and high income inequality that together makes the healthy food basket less affordable to weaker population groups. It should be noted that agricultural adaptation to climate change includes changes in the crop mix that may lead to price changes [39] in a way that can further exacerbate food security among the poor.

The current government’s import-exposure policy may contribute to lowering the cost of living and increasing food affordability in the short term, but it increases the country’s exposure to outside risk. Specifically, the reduction of tariffs on fruits and vegetables, which, despite the increase in their price in recent years, are still cheaper in Israel than in most developed countries [16], endangers local production capacity and exposes the Israeli consumer to greater future risk. Imports of fruits and vegetables from neighboring countries such as Turkey, Jordan, and Egypt may be attractive under the current conditions. However, the reliance on imports from these countries, which are expected to suffer more from climate change than Israel, may be problematic in the long term, not to mention the inherent risk of geopolitical developments in these countries and their relations with Israel, a risk that becomes more evident since the October 2023 war.

The war has exposed another internal risk though, which is that of labor shortage during periods of conflict. The combination of the internal risks of climate change and labor shortage and the external risks associated with food imports places the durability of the food supply in Israel at a problematic point. However, a risk-management strategy could minimize the threat to future food security. Maintaining the viability of local producers is vital to this goal, and the necessary market reforms should be implemented wisely and gradually so as to minimize uncertainty and suspicion of the farmers [40]. Agricultural R&D investments should be enlarged in order to enhance agricultural productivity and help the sector maintain the supply of fresh produce even as land and other resources continue to shrink. Specific attention should be devoted to developing labor-saving technologies in order to reduce the sector’s dependence on foreign workers. Agricultural-insurance programs should be expanded to cover not only weather damages but also those resulting from other events beyond the control of farmers [41,42].

The food security of specific population groups in Israel is affected not only by the availability and price of food, but also by their purchasing power. As Israel is one of the least equal countries in the developed world, it needs a policy focused on its weaker population groups in order to help them obtain a food basket that meets their needs. Moreover, it is necessary to strive for a food basket that will bring these population groups as close as possible to what is defined as a “healthy food basket”. Promoting health through healthy food is not only a private interest of each household, but also of society as a whole. In this context, the food-price-control policy, which currently includes many products that are not considered healthy, and the food-aid policy for needy families, which suffers from a rather low budget, must be reconsidered. Many families tend to consume unhealthy food not for economic reasons but due to lack of awareness or lack of understanding of the health consequences [43]. It follows that nutritional education and advocacy (including limiting the advertising of harmful food) should be an integral part of food policy. Economic incentives may also help in cases where education and advocacy are not effective enough. Such incentives may include, for example, taxation of harmful products and price controls of healthy products. These market interventions generally have negative efficiency effects, so their application should be carefully considered subject to a cost–benefit analysis.

There is no shortage of organizations in Israel that deal with food security, but it is necessary for a national body to be established with powers that will enable it to coordinate the activity, supervise the formulation of the strategy, break it down into goals and feasible policy measures, and supervise their implementation. As in the European Union [44], such a body should adopt a holistic approach that deals with all aspects of the food chain (farm to fork), from production in the field, through to processing and marketing, to household consumption. The establishment of such an authority should lead to the stabilization of the policy environment, ease the tension, and restore trust between farmers and the government. This will contribute to the resilience of the agricultural sector and enhance food security.

Food-security policy should be closely linked to the relevant body of research. Future research is particularly required to assess the optimal risk-minimizing portfolio of the local production of fresh agricultural products and their imports.

To summarize, the main conclusions of this paper are as follows:Israel’s food-security situation is not bad in international comparison, but this is not necessarily sustainable.Israel cannot supply all of its food needs, and reliance on imported food is likely to increase.As both local production and imports are subject to increasing risks, the formulation of a risk-management strategy is required.The farm sector should be stabilized and modernized, with labor-saving R&D investments and a stable policy environment.A special authority is needed to establish a long-term food-security strategy.

## Figures and Tables

**Figure 1 foods-13-00187-f001:**
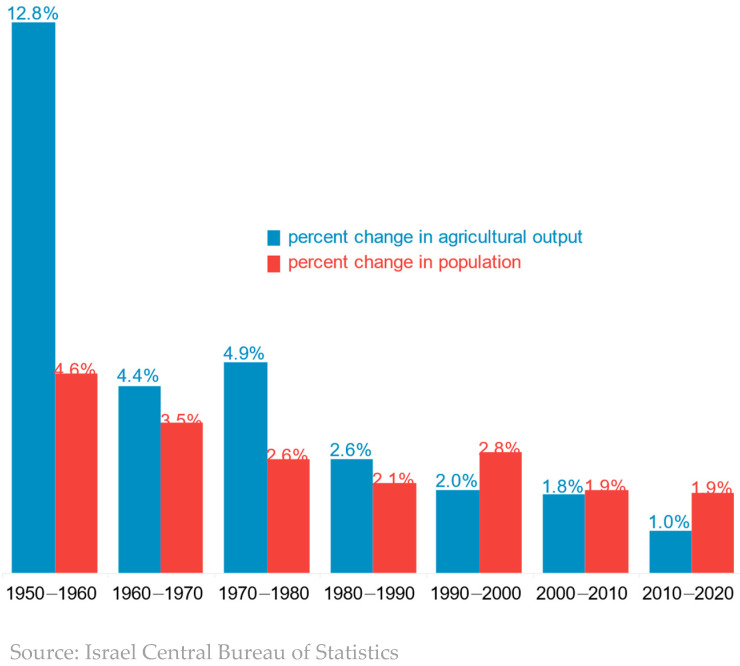
Annual increase in population and agricultural output.

**Figure 2 foods-13-00187-f002:**
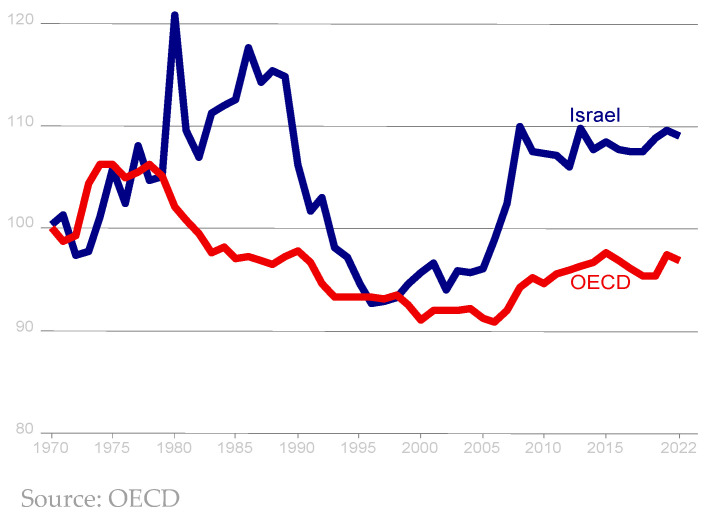
Real-food-price index, Israel and OECD average.

**Figure 3 foods-13-00187-f003:**
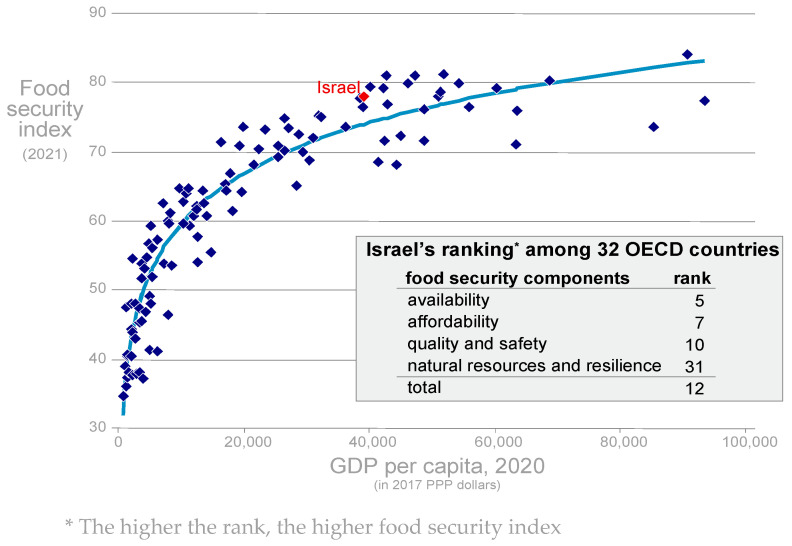
Food-security index by GDP per capita among 110 countries, 2021. This figure is adapted with permission from Ayal Kimhi (2022). Copyright 2022; Copyright Ayal Kimhi. Source: Economist Impact (2021).

**Figure 4 foods-13-00187-f004:**
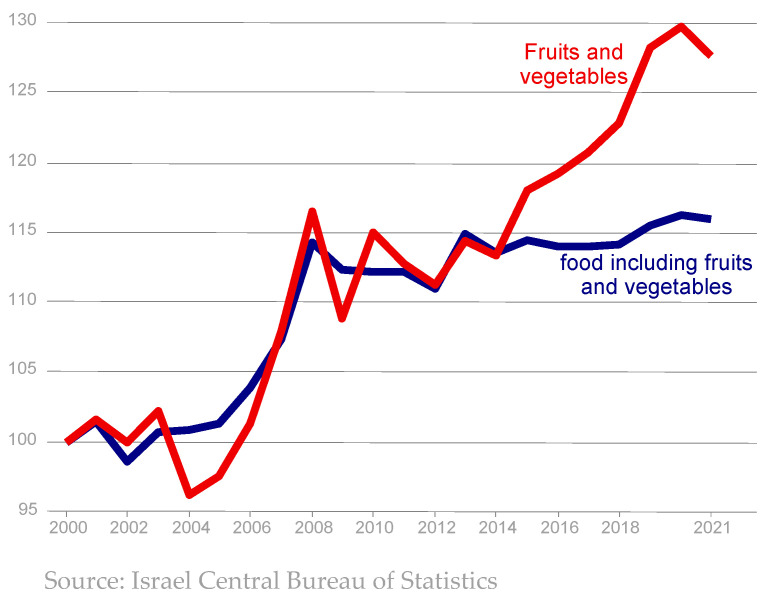
Real prices of food and of fruits and vegetables.

**Figure 5 foods-13-00187-f005:**
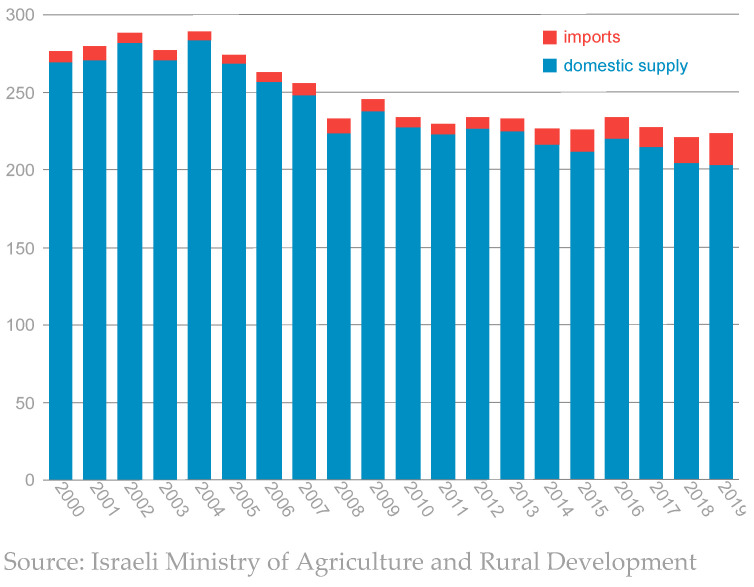
Supply of fruits and vegetables in Israel (annual kilograms per person).

**Figure 6 foods-13-00187-f006:**
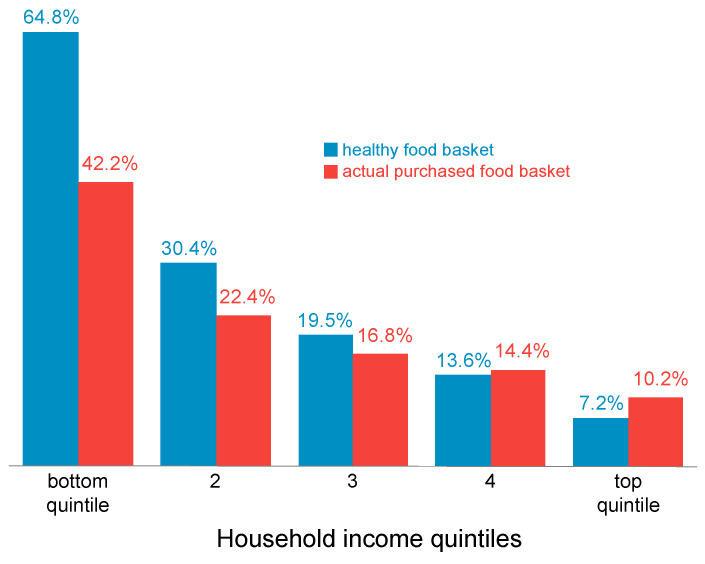
Expenditure on food as percent of household expenditure by income quintile, 2014. This figure is adapted with permission from Ayal Kimhi (2022). Copyright 2022; Copyright Ayal Kimhi.

**Figure 7 foods-13-00187-f007:**
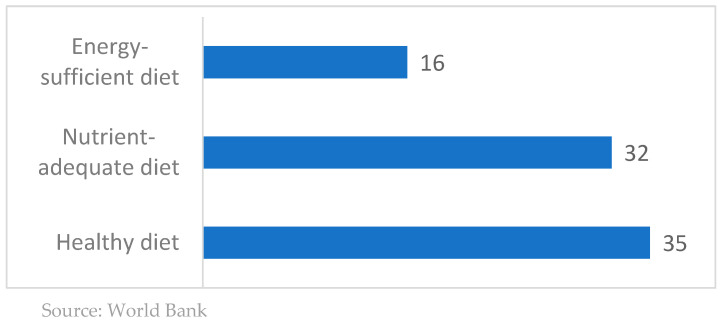
Israel’s rank in average daily cost of standard food baskets among 39 OECD countries (dollar cost per person per day of cheapest available food), 2017.

**Figure 8 foods-13-00187-f008:**
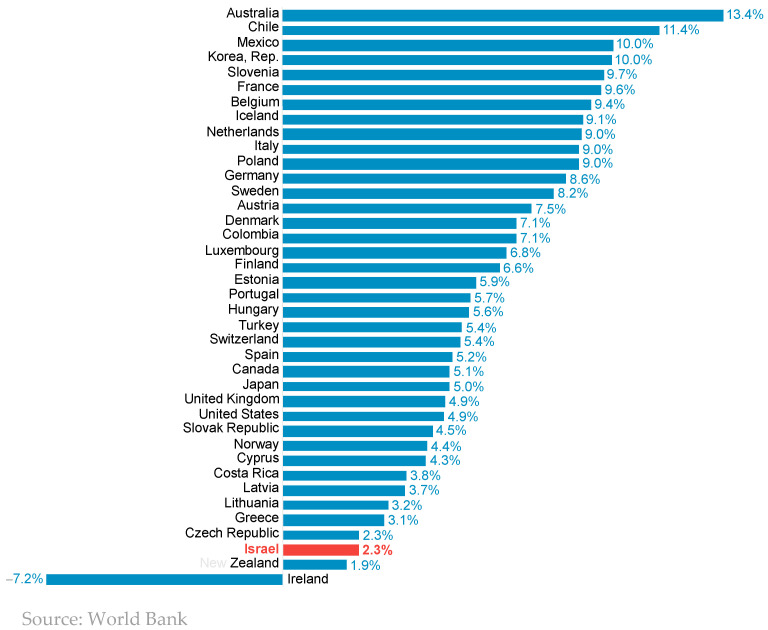
Changes in the cost of a healthy food basket between 2017 and 2020.

**Figure 10 foods-13-00187-f010:**
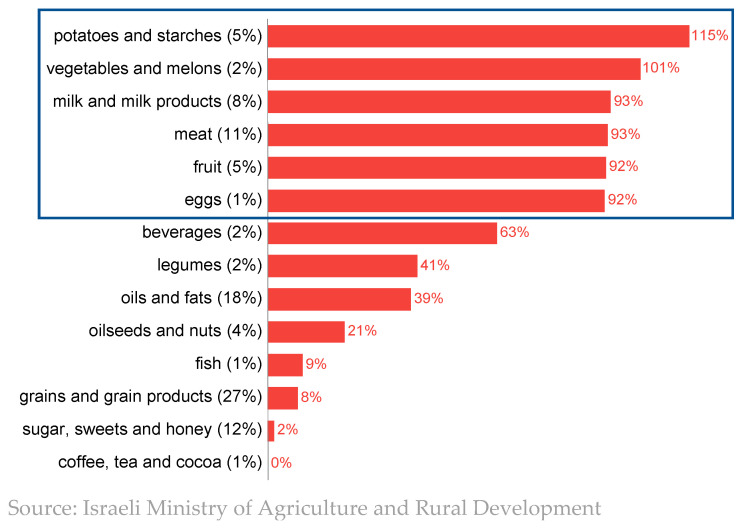
Share of food supplied domestically in 2020 (numbers in parentheses reflect the percent of total energy supply Items with more than 100% sufficiency implies that excess production is exported).

**Figure 11 foods-13-00187-f011:**
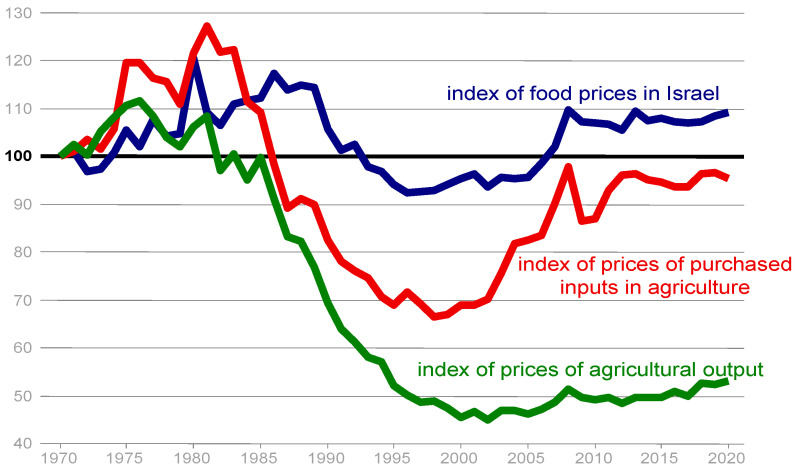
Real price indices (prices discounted by consumer price index) of food and agricultural inputs and output. Source: [31], updated. This figure is adapted with permission from Ayal Kimhi (2022). Copyright 2022; Copyright Ayal Kimhi; Source: Kislev and Zaban (2013), updated.

**Figure 12 foods-13-00187-f012:**
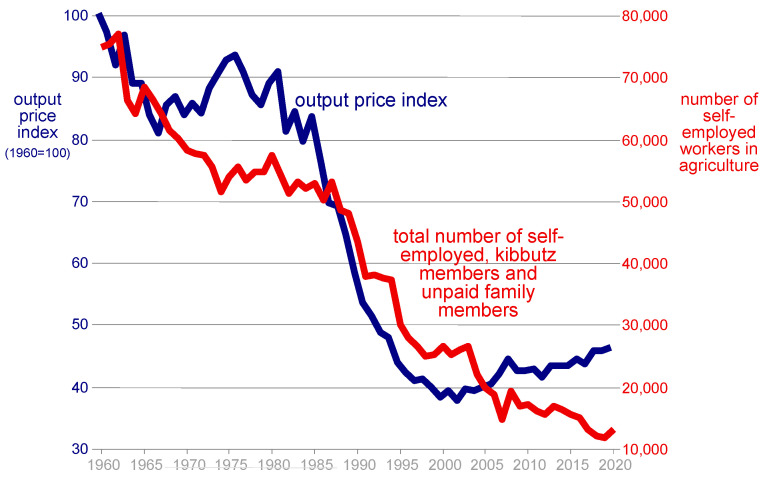
Output-price index and number of self-employed workers in agriculture. This figure is adapted with permission from Ayal Kimhi (2022). Copyright 2022; Copyright Ayal Kimhi; Source: Kislev and Zaban (2013), updated.

**Figure 13 foods-13-00187-f013:**
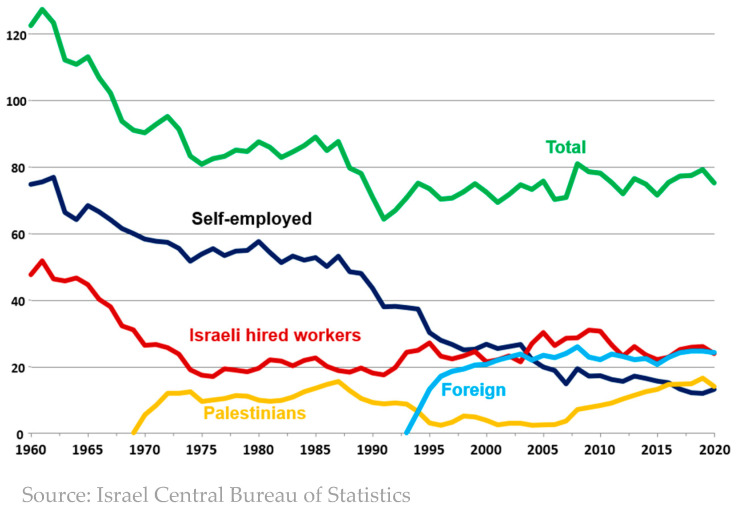
Workers in agriculture.

**Table 1 foods-13-00187-t001:** Israel’s ranking in the various components of the food-security index, 2021.

Food-Security Indicators ^1^	Weight	Israel’s Rank
**Availability**	**32.40%**	**5**
Sufficiency of supply	**26.30%**	**2**
Agricultural R&D	**9.10%**	**32**
Public expenditure on agricultural R&D	50.00%	15
Access to agricultural technology, education, and resources	50.00%	32
Agricultural infrastructure	**14.10%**	**24**
Road infrastructure	35.70%	24–30
Air, port, and rail infrastructure	35.70%	22
Irrigation infrastructure	28.50%	2
Volatility of agricultural production	**15.20%**	**7**
Political and social barriers to access	**12.10%**	**28**
Armed conflict	29.40%	26–29
Political stability risk	23.50%	27
Corruption	23.50%	19–28
Gender inequality	23.50%	19
Food loss	**14.10%**	**9**
Food-security and -access policy commitments	**9.10%**	**1–17**
Food-security strategy	50.00%	18–32
Food-security council	50.00%	1–3
**Affordability**	**32.40%**	**7**
Change in average food cost	**29.80%**	**2**
Proportion of population under global poverty line ^2^	**27.00%**	**20**
Inequality-adjusted income index ^3^	**29.80%**	**27**
Agricultural-import tariffs	**13.60%**	**24**
**Quality and safety**	**17.60%**	**10**
Dietary diversity	**20.30%**	**24**
Nutritional standards	**13.60%**	**13–21**
Micronutrient availability	**25.40%**	**2**
Protein quality	**23.70%**	**1**
Food safety	**16.90%**	**1–8**
**Natural resources and resilience**	**17.60%**	**31**
Exposure	**21.10%**	**9**
Temperature rise	27.30%	10
Drought	25.00%	23–32
Flooding	22.70%	3
Sea level rise	25.00%	14
Water	**14.00%**	**15–25**
Agricultural water risk—quantity	80.00%	15–32
Agricultural water risk—quality	20.00%	1–19
Land	**14.00%**	**7**
Land degradation	60.00%	6
Grassland	20.00%	1
Forest change	20.00%	29
Oceans, rivers, and lakes	**12.30%**	**5**
Eutrophication	50.00%	4–32
Marine biodiversity	50.00%	3
Sensitivity	**10.50%**	**32**
Food-import dependency	60.00%	32
Dependence on natural capital	40.00%	13
Political commitment to adaptation	**21.10%**	**31**
Demographic stress	**7.00%**	**32**
Projected population growth	75.00%	32
Urban absorbment capacity	25.00%	31

^1^ Primary components in bold; secondary components in black; and secondary subcomponents in gray. The sum of the weights of the components in each group or subgroup of components adds up to 100%. Source: [10]. ^2^ The proportion of the population whose daily income is less than $3.20 per day (at 2011 exchange rates adjusted for purchasing-power parity. ^3^ GNI per capita at 2011 PPP adjusted for level of inequality [11].

## Data Availability

The data presented in this study are available on request from the corresponding author. The data are not publicly available due to the fact that it was collected from many different sources.

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
