# Peer review of "Food Security in Israel: Challenges and Policies"

_foods, 2024, doi:10.3390/foods13020187_

Round 1

Reviewer 1 Report

Comments and Suggestions for Authors

Food security issue is always crucial for the development of societies and sustainable development, and this creates the rationale for this elaboration.

Furthermore, the black swans of our times have threaten the state of food and nutrition security all over the world.

1. This definition: "The internationally accepted definition of food security is that all people, at all times, have physical, social, and economic access to sufficient, safe, and nutritious food that meets their food preferences and dietary needs for an active and healthy life" needs a citation.

The data showed in Figure 1 is very interesting but there is a lack of explanation on what were the major causes of reducing agricultural output in Israel. Please, write a few sentences about it.

Figure 3 is also interesting but I think that Global Food Security index was developed by Economist Impact. See more: Kowalski, J., & Kowalska, A. (2022). The realization of the human right to food: preliminary remarks on assessing food security. PrzeglÄ…d Prawno-Ekonomiczny, 1, 9-31. https://doi.org/10.31743/ppe.13009.

The source (UNICEF) is not correct for Figure 3 and Table 1. I think that Chat GPT might be one of the authors of the manuscript.

My another question is regarding this source: Central Bureau of Statistics. We do not know if it is Israeli Bureau or some other one.

There is not enough information on "healthy food basket". This concept needs to be explained in detail. It has been done before for Kenya: https://www.frontiersin.org/articles/10.3389/fsufs.2023.1181683/full

I am not sure if this sentence is right: This implies that the problem of the weaker households who cannot afford a healthy food basket is not a problem of price increases but of purchasing power. Food inflation rate has been high everywhere over the past 2 years or so.

I do not see references for the source of data in Figure 7 and 8, ie. World Bank.

The authors write: Today, the list of price-controlled products includes 273 bread, salt, dairy products and eggs (Azarieva and others, 2016). Is it still correct? The citation is quite old (2016).

I do not agree with that "The price 2of salt, another unhealthy food, is also controlled." We need a certain amount of salt. It is not definitely unhealthy. Scientific paper should always consider threats and benefits.

I do not agree with this narrative either: However, the milk products whose prices are controlled include butter, cream and hard cheeses which are high in fat. In general, it can be said that the food price control policy does not coincide with the promotion of the consumption of a healthy food basket.

Cheeses and butter are also beneficial for human health, especially children health.

Part 2.3 is too short. The analysis should be more thorough. There is not enough text about natural resources and sustainability issues influencing the state of food security in Israel.

The newest data in Figure 11 are needed. If the authors wirte about the Covid-19 pandemic, the war in Ukraine and Israel, they need to analyse the data as new as possible.

The discussion part misses the relevant references. The citation of an article published in 2003 is not relevant. I still think that Chat GPT was a key co-author.

There is no conclusion. The paper must be concluded.

Anyway, the paper is interesting but it needs to be revised thoroughly.

There is a lack of the main aim of the study and methodological approach part.

Comments on the Quality of English Language

The quality of Enslish language is quite good.

Author Response

Thanks for your comments and suggestions. I have collected them in the attached file and added responses.

Reviewer 2 Report

Comments and Suggestions for Authors

Dear Authors

I showed my suggestions on the text

With my compliments and best regards.

Author Response

Thanks for your comments and suggestions. I have responded to each of them in the attached file.

Round 2

Reviewer 1 Report

Comments and Suggestions for Authors

The author(s) revised the manuscript.

The most important thing now is to think about methodological approach used in this study and to wirte about it in the manussript. The scholarly journals publish scintific articles where the methods are described (even very shortly).

This part needs a reference:  A healthy diet provides not only adequate calories but also adequate levels of all essential nutrients and food groups needed for an active and healthy life. The cost of a healthy diet is defined as the cost of the least expensive locally available foods to meet requirements for energy and food-based dietary guidelines for a representative person within energy balance at 2 330 kcal/day. The guidelines explicitly recommend food quantities for each food group and provide a wide regional representation.

Author Response

I have added a methodology paragraph and the reference